# Clusters in Infant Environmental Factors Influence School-Age Children’s Vegetable Preferences in Japan

**DOI:** 10.3390/nu16071080

**Published:** 2024-04-06

**Authors:** Yudai Yonezawa, Tomoka Okame, Nozomi Tobiishi, Yume Tetsuno, Miho Sakurai, Shigenori Suzuki, Yuji Wada

**Affiliations:** 1Diet & Well-Being Research Institute, KAGOME Co., Ltd., 17 Nishitomiyama, Nasushiobara 329-2762, Japanshigenori_suzuki@kagome.co.jp (S.S.); 2Tokyo Head Office, KAGOME Co., Ltd., 3 Chome-21-1 Nihonbashihamacho, Tokyo 103-0007, Japan; 3College of Gastronomy Management, Ritsumeikan University, 1 Chome-1-1 Nojihigashi, Kusatsu 525-0058, Japanyujiwd@fc.ritsumei.ac.jp (Y.W.); 4Graduate School of Gastronomy Management, Ritsumeikan University, 1 Chome-1-1 Nojihigashi, Kusatsu 525-0058, Japan

**Keywords:** childcare, environmental factors in infancy, nutrition education, Japan, vegetable preferences

## Abstract

It remains unclear how the various environmental factors are combined in practice to influence vegetable preferences in school-aged children. This study aimed to clarify the environmental factors during infancy and their association with vegetable preference in school-aged children. To find clusters of early childhood environmental factors, we conducted a factor analysis on 58 items related to early childhood environmental factors and a k-means cluster analysis using the factors obtained. The association of the extracted factors and clusters with vegetable preferences was assessed by multiple regression analysis. Twelve factors relating to vegetable eating, cooking and harvesting experience, and parental attitudes were extracted by factor analysis. Three clusters, “low awareness of experiences”, “high awareness” and “low positive encouragement”, were then extracted. In the multiple regression analysis, all 12 factors were found to be associated with vegetable preferences. Furthermore, it was found that the “high awareness” group had a significantly higher score for vegetable preference than the “low awareness of experiences” group (β = 0.56, 95% CI: 0.37–0.74). Thus, the study found that environmental factors during infancy, in isolation and combination, influenced vegetable preferences in school-aged children. Assessing the combination of various environmental factors during infancy may contribute to a better understanding of future vegetable preferences.

## 1. Introduction

Vegetables are a healthy food group, low in calories yet rich in vitamins, minerals, and fiber [1]. Studies have shown that increasing vegetable consumption is associated with the prevention of obesity and is related to alleviating chronic diseases [2]; furthermore, regular vegetable consumption may also contribute to long-term positive health in individuals. However, average vegetable consumption in Japan remains below the recommended amount of 350 g/day [3]. Moreover, a systematic review of vegetable consumption and recommendations reported that vegetable consumption in many countries did not reach recommended levels [4]. Thus, increasing vegetable consumption is a pressing issue for achieving better levels of well-being among the world population.

Numerous previous studies have indicated that environmental factors during infancy affect vegetable intake and preferences later in life [5,6]. A recent umbrella review of several systematic reviews summarizing previous studies on strategies to promote vegetable liking in the early years of life found that repeated exposure to a single or a variety of vegetables was promising evidence [5]. The umbrella review also reported that several strategies, like increasing familiarity with vegetable flavors and/or willingness to try vegetables, are emerging evidence to improve liking toward vegetables [5]. Furthermore, environmental factors in preschools and homes work together in practice; hence, it is important to know how these combinations work better. A cross-sectional study conducted in the US classified parenting practices using cluster analysis and reviewed the association between these clusters and vegetable intake in preschoolers [7]. However, studies assessing the combined effects of complex environmental factors on vegetable preferences are scarce.

Thus, this study aimed to find clusters of early childhood environmental factors related to the development of a preference for vegetables in Japan. We conducted a large-scale web-based questionnaire survey of Japanese households and analyzed the association between environmental factors during infancy and vegetable preferences at school age. Thereafter, we used cluster analysis to classify environmental factors during infancy and evaluated how these clusters influenced vegetable preferences in school-age children.

## 2. Materials and Methods

### 2.1. Study Design and Population

In November 2021, web-based research was conducted with 1500 mothers with children among the panel of Japanese participants at ASMARQ Co., Ltd., Shibuya-ku, Japan. The data that support this study’s findings are available from the corresponding author upon reasonable request. To target the content implementable in practice nursery schools and households, original question items on the environmental factor at ages 0–5 years were developed in this study instead of using existing questionnaires. The question items were developed through group work using the KJ-method [8] by the seven researchers of this study, based on the initiatives of the nursery school implementing the experiential curriculum to develop vegetable preferences. The KJ-method was performed online using Google Slides. First, the initiatives involved in the curriculum were verbalized and categorized by the aims of childcare. Thereafter, hypotheses were formulated to examine the kind of environmental factor (e.g., people, facilities, childcare curriculum, etc.) in the nursery school that makes these initiatives possible. The hypotheses were rearranged and categorized using the KJ-method, comparing them with lives and nurturing environmental factors in households. Each extracted group was assigned a name, and the elements that composed these groups were listed, resulting in 58 question items related to the environmental factor, including food, inhabitants, encouragement, and experiences in children aged 0–5 years. The final version of the questionnaire included 58 questions about environmental factors at ages 0–5 years, vegetable preferences at first grade, and other information items (e.g., basic characteristics of mother and child, family structure, and other food preferences). Among the participants, six who indicated that they did not enroll their children in childcare facilities were excluded, resulting in a total of 1494 participants for the final analysis.

### 2.2. Variables

A total of 58 question items about the environmental factor at ages 0–5 years were rated on a 5-point Likert scale (with the following response options: not applicable, low applicable, fairly applicable, good applicable, and highly applicable), all of which are listed in Appendix A. Preferences for vegetables in the first grade of elementary school were assessed on a 7-point Likert scale (with the following response options: strongly dislike, dislike, somewhat dislike, neither like nor dislike, somewhat like, like, and strongly like). Mothers’ ages were categorized into four age groups (<25, 25–29, 30–34, and ≥35 years). Residential areas were divided into eight categories (Hokkaido, Tohoku, Kanto, Chubu, Kinki, Chugoku, Shikoku, and Kyushu/Okinawa). Family members living together were classified into four categories (no spouse and no grandparents, spouse and no grandparents, no spouse and grandparents, and spouse and grandparents). The number of children was categorized into two categories (one, two or more). Employment status was classified into two categories (yes and no); educational attainment was divided into three categories (junior high or high school graduate, junior college or vocational school graduate, and college or graduate school graduate). Children’s ages were divided into two categories (six and seven years). The sex of the child was classified into three categories (male, female, and no answer). The presence or absence of food allergies in children was divided into two categories (yes and no). The variables related to the child’s anger at five years old (when the child was five years old, the child showed intense emotions when not given what they wanted) and curiosity at five years old (when the child was five years old, the child was curious about new things) were evaluated using five categories (not applicable, low applicable, fairly applicable, good applicable, and highly applicable).

### 2.3. Factors Related to Environmental Factors during Infancy

An exploratory factor analysis (EFA) was conducted using 58 questions about environmental factors at ages 0–5 years to extract factors related to environmental factors during infancy. The analysis involving EFA was performed using the psych package [9]. Data validity was assessed by the Kaiser-Meyer-Olkin (KMO) measure of sample validity, which was considered to be acceptable when this value was greater than 0.60. The KMO was calculated using the KMO function [9]. The scree plot, Velicer’s minimum average partial (MAP) test and Bayesian Information Criterion (BIC) were used to determine the optimal number of factors. The scree plot was created using the fa.parallel function, while the MAP test and BIC were performed using the vss function [9]. A total of 20 analyses were conducted, including five patterns for the number of factors (8–12), two for the factor extraction method (unweighted least squares (MINRES) and maximum likelihood (ML)), and two rotation methods for the factor axes (orthogonal rotation (varimax) and diagonal rotation (promax)). EFAs were conducted using the fa function [9]. The optimal analysis conditions were selected based on statistical criteria and possible interpretations of the factors. Each factor was named by three different researchers based on the questionnaire items and their factor loading values contributing to each factor.

### 2.4. Cluster Analysis Using Factors Related to Environmental Factors during Infancy

A k-means cluster analysis was conducted using the factor scores of the factors obtained by the EFA. The analysis was performed using the kmeans function of the ClusterR package [10]. The k-means function was used to set the number of clusters to 3–5, and the optimal analysis conditions were selected based on the analysis results from CLUSPLOT analysis [11] and the possible interpretation of each cluster. Each cluster was named by three different researchers based on the factor loading values of each factor in each cluster.

### 2.5. Statistical Analysis

Multiple linear regression analysis was utilized to analyze the association between the environmental factor in infancy and vegetable preferences in school-aged children. Factor scores of environmental factors at age 0–5 years extracted by factor analysis (continuous values) or clusters extracted by k-means (categorical values) were used as the exposure and total vegetable preferences (continuous values) at first grade as the outcome. Multivariable model 1 used the mothers’ age, residential area, family members living together, number of children, employment status, educational attainment, age of children, sex, and food allergy status as confounders. In multivariable model 2, along with the factors used in multivariable model 1, we also used the child’s proneness to anger and their curiosity at age five as confounders. Confounders were determined by listing potential confounding factors from previous studies and our findings and scrutinizing whether they matched the three criteria for confounders [12]. Continuous variables are presented as means and standard deviations (SDs), and categorical variables as frequencies and percentages. R (Version 4.1.2) was used as the statistical analysis software.

## 3. Results

### 3.1. Characteristics of the Study Population

Table 1 shows the characteristics of the study participants. The most common category of mothers aged 35 years or older and the most common regions of residence were Kanto, Kinki, and Chubu, in that order. More than 80% of the participants lived with their spouses, but not with their grandparents. Similarly, in over 80% of the participants’ families, the children had siblings. The male-to-female ratio of children was slightly higher for males, but roughly 1:1.

### 3.2. Exploratory Factor Analysis of Environmental Factors of Children Aged 0–5 Years

An EFA was conducted using 58 questions about the environmental factors of infancy for factor analysis (Appendix A). The scree plot is shown in Figure 1. Based on the statistical criteria and the possibility of interpreting the factors, it was determined that the optimal conditions for analysis were 12 factors and a promax rotation using the MINRES method. In the process of factor analysis, six items with low factor loadings were excluded. The obtained factors were “vegetable growing and harvesting experience”, “environment in which cooking is experienced daily through visual, auditory, and olfactory senses”, “balanced meals with vegetable awareness”, “opportunities to be involved in cooking from the stage of vegetable preparation”, “positive family attitudes during meals”, “nutrition education experience at preschool”, “positive attitudes in daily life”, “positive attitudes toward children when helping”, “experience playing with vegetables in situations other than eating”, “experience in tasting vegetables”, “positive encouragement of children when cooking”, and “negative attitudes toward children when cooking”.

### 3.3. Associations between Factor Scores of 12 Factors Related to Environmental Factors at Age 0–5 Years and Vegetable Preferences

Multiple regression analysis was performed to analyze the association between factor scores related to the child’s environmental factor at ages 0–5 years and vegetable preferences at first grade (Table 2). Consequently, all 12 factors were found to be associated with vegetable preferences. Especially in multivariable model 1, high regression coefficients were observed for “experience in tasting vegetables”, “balanced meals with vegetable awareness”, and “opportunities to be involved in cooking from the stage of vegetable preparation”. Similar results were also obtained for multivariable model 2.

### 3.4. Cluster Analysis of Factors Related to Environmental Factors at Age 0–5

A k-means cluster analysis was conducted using factor scores for factors related to environmental factors at ages 0–5 to create groups of participants with similar characteristics (Table 3). From the CLUSPLOT analysis and the possibility of interpreting each cluster, the result of three clusters was selected as the optimal condition for the analysis. Consequently, the following groups were extracted: “low awareness of experiences”, “high awareness”, and “low positive encouragement”. “Low awareness of experiences” was found to be a cluster with low factor scores for factors related to experience such as “vegetable growing and harvesting experience”, “experience playing with vegetables in situations other than eating”, and “experience in tasting vegetables”. “High awareness” is a cluster where factor scores for positive factors are generally high and only the negative factor, “negative attitudes toward children when cooking”, is low. Finally, “low positive encouragement” is a cluster with low factor scores for positive attitudes, like “positive family attitudes during meals” and “positive attitudes toward children when helping”, and high scores for negative attitudes, like “negative attitudes toward children when cooking”.

### 3.5. Association between Each Cluster and Vegetable Preference

Multiple regression analysis was conducted to identify the relationship between each cluster and vegetable preference (Table 4). Based on the results, in all models, including multivariable model 1, and multivariable model 2, the scores of vegetable preference were significantly higher in the “high awareness” category compared to the “low awareness of experiences”. Contrarily, in all models, no significant difference in vegetable preference scores was observed between the “low awareness of experiences” and the “low positive encouragement”.

The present study found that all 12 factors related to environmental factors at ages 0–5 obtained by factor analysis were related to vegetable preferences in the first grade of elementary school. “Experience in tasting vegetables”, “balanced meals with vegetable awareness”, and “opportunities to be involved in cooking from the stage of vegetable preparation” had a particularly large impact. Furthermore, the results of k-means cluster analysis and multiple regression analysis using the factor scores showed that “high awareness” had significantly higher scores of children’s vegetable preference compared to “low awareness of experiences”. Although research on the combined effects of environmental factors during infancy on vegetable preferences is limited, the present study provides new insights into these.

Cluster analysis of factors related to environmental factors at ages 0–5 extracted “low awareness of experiences”, “high awareness”, and “low positive encouragement”. Notably, it was found that those with high awareness of child-rearing were concerned about a wide range of environmental factors in raising their children. Furthermore, the higher points for vegetable preference in this group suggested that improving a wide range of environmental factors would have created a positive effect on vegetable preference. A complex combination of environmental factors, like exposure to taste, the provision of dishes containing a variety of vegetables at home, and experience cooking with vegetables, could be inferred to improve vegetable preferences in children. Previous studies suggest that environmental factors involving these elements have a positive impact on vegetable intake and preferences [5,6,13,14,15]. However, the generalizability and applicability of the results to other regions with different diets, lifestyles and degrees of nutrition education are limited because the present study was conducted with Japanese participants. Further complex studies in countries around the world are warranted to reveal in detail how environmental factors combine and function in practice. In particular, it is necessary to verify whether the implementation of the present intervention on complex environmental factors at home and in nurseries will improve children’s preference for vegetables.

Focusing on individual factors obtained by factor analysis, “experience in tasting vegetables” had the highest regression coefficient for vegetable preferences. An umbrella review of strategies to enhance vegetable preferences in the early years of life reported that “repeated exposure to a single or a variety of vegetables” was promising evidence [5]; the current study supports these previous results. Furthermore, “increasing familiarity with vegetable flavors” and “willingness to try vegetables” were reported as emerging evidence [5], consistent with multiple factors affecting vegetable preferences in the present study. For example, “increasing familiarity with vegetable flavors” related factors such as “balanced meals with vegetable awareness” and “opportunities to be involved in cooking from the stage of vegetable preparation” and “willingness to try vegetables” related factors such as “positive family attitudes during meals” and “positive attitude toward children when helping” were in the top half of the 12 factors with regression coefficients for vegetable preferences. Practical implementation of these methods in preschools and households is encouraged. For instance, it may be helpful to address these issues comprehensively. This can be achieved by providing opportunities for communal cooking of vegetables and other foods at home and in nurseries. Another approach is to replace snacks with boiled or steamed vegetables.

This study has several limitations. First, data collection was conducted using an Internet survey, which may have biased the demographics of the participants. Second, there is a possibility of sample error problems because the significant sampling method of monitor enrolment was employed. Third, there may be a discrepancy with the actual vegetable preferences of children, as questions about the same were asked of their mothers. Fourth, recall bias may have arisen because the questionnaire included questions about past events. In addition, the questionnaire developed in the present study has not been validated. In the present study, parents with children who liked vegetables may have responded more positively to the question on environmental factors in infancy, which may have influenced the conclusions.

## 4. Conclusions

It was found that all 12 factors related to the environmental factor at ages 0–5 were associated with children’s vegetable preferences. Of these, factors related to “experience in tasting vegetables”, “balanced meals with vegetable awareness”, and “opportunities to be involved in cooking from the stage of vegetable preparation” had a particularly large impact. Furthermore, the k-means method identified the “high awareness” cluster with generally high scores on positive factors and low scores on negative factors, and participants belonging to this cluster had higher scores for their children’s vegetable preferences. Thus, combining and implementing approaches from infancy may help to improve vegetable preferences in school-age children.

## Figures and Tables

**Figure 1 nutrients-16-01080-f001:**
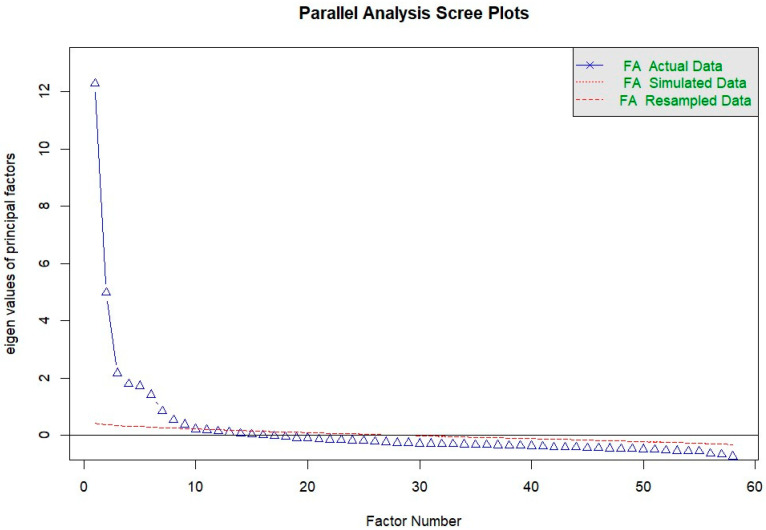
Scree plot with questions on environmental factors in infancy. FA—factor analysis.

**Table 1 nutrients-16-01080-t001:** Characteristics of mothers and their children.

	All Participants
	(*n* = 1494)
Mother	
Age (in years)	
<25	67 (4.5)
25–29	312 (20.9)
30–34	504 (33.7)
≥35	611 (40.9)
Residential area	
Hokkaido	73 (4.9)
Tohoku	82 (5.5)
Kanto	551 (36.9)
Chubu	219 (14.7)
Kinki	289 (19.3)
Chugoku	89 (6.0)
Shikoku	31 (2.1)
Kyusyu	160 (10.7)
Family members living together	
No spouse and no grandparents	90 (6.0)
Spouse and no grandparents	1254 (83.9)
No spouse and grandparents	38 (2.5)
Spouse and grandparents	112 (7.5)
The number of children	
One	295 (19.7)
Two or more	1199 (80.3)
Employment status	
Yes	941 (63.0)
No	553 (37.0)
Educational attainment	
High school graduate or less	443 (29.7)
Junior college or vocational college graduate	493 (33.0)
University graduate or above	558 (37.3)
Children	
Age (years)	
6	518 (34.7)
7	976 (65.3)
Sex	
Male	805 (53.9)
Female	687 (46.0)
No answer	2 (0.1)
Food allergies	
Yes	109 (7.3)
No	1385 (92.7)
Data are *n* (%).	

**Table 2 nutrients-16-01080-t002:** Associations between factor scores of 12 factors related to environmental factors at age 0–5 years and vegetable preferences.

Factors				Clude				Multivariable Model 1 ^a^				Multivariable Model 2 ^b^
	β	95% CI		*p* Value	β	95% CI		*p* Value	β	95% CI		*p* Value
		Lower	Upper			Lower	Upper			Lower	Upper	
Vegetable growing and harvesting experience	0.19	0.10	0.28	<0.001	0.24	0.16	0.32	<0.001	0.21	0.13	0.29	<0.001
Environment in which cooking is experienced daily through visual, auditory, and olfactory senses	0.23	0.15	0.30	<0.001	0.18	0.10	0.26	<0.001	0.14	0.06	0.22	<0.001
Balanced meals with vegetable awareness	0.46	0.38	0.54	<0.001	0.49	0.41	0.57	<0.001	0.45	0.37	0.53	<0.001
Opportunities to be involved in cooking from the stage of vegetable preparation	0.47	0.39	0.55	<0.001	0.46	0.38	0.54	<0.001	0.43	0.35	0.51	<0.001
Positive family attitudes during meals	0.26	0.18	0.34	<0.001	0.28	0.19	0.36	<0.001	0.23	0.14	0.31	<0.001
Nutrition education experience at preschool	0.12	0.04	0.21	0.006	0.11	0.02	0.19	0.017	0.05	−0.04	0.14	0.274
Positive attitude in daily life	0.18	0.10	0.27	<0.001	0.18	0.09	0.27	<0.001	0.17	0.08	0.26	<0.001
Positive attitude toward children when helping	0.60	0.52	0.69	<0.001	0.27	0.18	0.35	<0.001	0.23	0.14	0.31	<0.001
Experience playing with vegetables in situations other than eating	0.30	0.21	0.38	<0.001	0.29	0.20	0.37	<0.001	0.26	0.18	0.35	<0.001
Experience in tasting vegetables	0.21	0.13	0.30	<0.001	0.64	0.56	0.72	<0.001	0.61	0.53	0.70	<0.001
Positive encouragement of children when cooking	0.15	0.07	0.23	<0.001	0.19	0.10	0.28	<0.001	0.18	0.09	0.26	<0.001
Negative attitude toward children when cooking	−0.09	−0.18	0.01	0.071	−0.20	−0.29	−0.11	<0.001	−0.13	−0.23	−0.04	0.004

^a^ Adjusted for age of mothers, residential area, family members living together, number of children, employment status, educational attainment, age of children, gender, and food allergy status. ^b^ Adjusted for confounders in multivariable model 1 + the child’s anger proneness at age 5 and the child’s curiosity at age 5. 95% CI 95% confidence interval, β regression coefficient.

**Table 3 nutrients-16-01080-t003:** Cluster analysis of factors related to environmental factors at age 0–5.

	Cluster 1	Cluster 2	Cluster 3
	Low Awareness of Experiences	High Awareness	Low Positive Encouragement
	(*n* = 587)	(*n* = 539)	(*n* = 368)
Factor scores for factors related to environmental factors during infancy			
Vegetable growing and harvesting experience	−0.55 ± 0.76	0.74 ± 0.85	−0.22 ± 0.75
Environment in which cooking is experienced daily through visual, auditory, and olfactory senses	0.35 ± 0.52	0.41 ± 0.60	−1.17 ± 1.02
Balanced meals with vegetable awareness	−0.08 ± 0.75	0.68 ± 0.65	−0.87 ± 0.85
Opportunities to be involved in cooking from the stage of vegetable preparation	−0.39 ± 0.76	0.80 ± 0.67	−0.55 ± 0.76
Positive family attitudes during meals	0.21 ± 0.58	0.56 ± 0.63	−1.15 ± 0.77
Nutrition education experience at preschool	−0.16 ± 0.81	0.55 ± 0.72	−0.55 ± 0.87
Positive attitude in daily life	−0.29 ± 0.89	0.43 ± 0.82	−0.18 ± 0.76
Positive attitude toward children when helping	0.03 ± 0.74	0.64 ± 0.59	−0.98 ± 0.69
Experience playing with vegetables in situations other than eating	−0.51 ± 0.70	0.70 ± 0.69	−0.20 ± 0.78
Experience in tasting vegetables	−0.42 ± 0.74	0.71 ± 0.65	−0.36 ± 0.75
Positive encouragement of children while cooking	−0.52 ± 0.80	0.45 ± 0.70	0.17 ± 0.75
Negative attitude toward children while cooking	−0.17 ± 0.69	−0.34 ± 0.85	0.76 ± 0.68

Data are Mean ± SD. SD standard deviation.

**Table 4 nutrients-16-01080-t004:** Association between each cluster and vegetable preference.

				Clude				Multivariable Model 1 ^a^				Multivariable Model 2 ^b^
	β	95% CI		*p* Value	β	95% CI		*p* Value	β	95% CI		*p* Value
		Lower	Upper			Lower	Upper			Lower	Upper	
Cluster 1	Reference				Reference				Reference			
Low awareness of experiences												
Cluster 2	0.65	0.47	0.82	<0.001	0.56	0.37	0.74	<0.001	0.49	0.31	0.68	<0.001
High awareness												
Cluster 3	−0.11	−0.31	0.08	0.26	0.00	−0.21	0.21	0.99	0.05	−0.16	0.25	0.66
Low positive encouragement												

^a^ Adjusted for age of mothers, residential area, family members living together, number of children, employment status, educational attainment, age of children, gender, and food allergy status. ^b^ Adjusted for confounders in multivariable model1 + the child’s anger proneness at age 5 and the child’s curiosity at age 5. 95% CI 95% confidence interval, β regression coefficient.

## Data Availability

The original contributions presented in the study are included in the article/Appendix A, further inquiries can be directed to the corresponding authors.

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
