# Peer review of "Clusters in Infant Environmental Factors Influence School-Age Children’s Vegetable Preferences in Japan"

_nutrients, 2024, doi:10.3390/nu16071080_

Round 1

Reviewer 1 Report

Comments and Suggestions for Authors

This article provides important insights into how environmental factors during infancy impact children's vegetable preferences. The study is well-designed, the methods are sound, and the data analysis is thorough, making a valuable contribution to the fields of nutrition and public health. I would suggest further improvements in the following areas:

1. The study focuses on Japanese children, and its results may be influenced by cultural differences. I recommend that the authors discuss the generalizability of these findings and their applicability in different cultural contexts.

2. The use of a non-validated questionnaire requires more cautious interpretation of the results. I would advise the authors to not only acknowledge this limitation in the discussion section but also explore its potential impact on the study's conclusions.

3. It is advisable for the authors to propose directions for future research in the discussion section, particularly regarding how to further validate and expand upon the current findings.

4. The authors adjusted for several confounding factors in the model. I suggest that the authors provide detailed explanations for the selection of each confounding factor and discuss their potential biological mechanisms related to vegetable preferences.

5. The authors utilized k-means cluster analysis to identify clusters of environmental factors. I recommend that the authors explore the robustness of the clustering results, for instance, by validating the results using different numbers or methods of clustering (such as hierarchical clustering).

6. The study results indicate that increasing awareness of environmental factors during infancy may help improve children's vegetable preferences. I recommend that the authors provide specific, actionable intervention recommendations in the discussion section so that practitioners can apply these findings.

7. To better illustrate the research findings, I suggest that the authors include more charts and visual elements in the article to help readers understand the complex data analysis and clustering results.

Reviewer 2 Report

Comments and Suggestions for Authors

This study aimed to identify "clusters of early childhood environmental factors related to the development of preference for vegetables in Japan".

1. The text presenting cluster analysis and factor analysis should present: a) references and b) tools (not only the language, but also libraries, frameworks, etc.)

2. Section 2.5. Statistical Analysis presents two models using different features/variables, excluding or adding ones and merging others. What is the rationale to use them?

3. A scree plot presenting the factor loadings should be added.

4. Make the answers to the following questions clear in the manuscript, possibly in the discussion. This will increase the number of related works cited, which is low.

a. What is the novelty of the study?

b. What are its contributions?

c. Are the findings not already well known today?

Round 2

Reviewer 2 Report

Comments and Suggestions for Authors

The authors answered my questions and addressed my concerns.